# A Dual Biomarker TK1 Protein and CA125 or HE4-Based Algorithm as a Better Diagnostic Tool than ROMA Index in Early Detection of Ovarian Cancer

**DOI:** 10.3390/cancers15051593

**Published:** 2023-03-03

**Authors:** Diana Cviič, Kiran Jagarlamudi, Leon Meglič, Erik Škof, Andrej Zore, David Lukanović, Staffan Eriksson, Joško Osredkar

**Affiliations:** 1Institute of Clinical Chemistry and Biochemistry, University Medical Centre Ljubljana, Zaloška c. 002, 1000 Ljubljana, Slovenia; 2Research and Development Division, AroCell AB, 11151 Stockholm, Sweden; 3Division of Gynecology, Department of Gynecology, University Medical Centre Ljubljana, Zaloška c. 002, 1000 Ljubljana, Slovenia; 4Medical Faculty, University Ljubljana, Vrazov trg 1, 1000 Ljubljana, Slovenia; 5Institute of Oncology, Zaloška c. 002, 1000 Ljubljana, Slovenia; 6Department of Anatomy, Physiology & Biochemistry, Swedish University of Agricultural Science, VHC, 75007 Uppsala, Sweden; 7Faculty of Pharmacy, University of Ljubljana, Aškerčeva c. 7, 1000 Ljubljana, Slovenia

**Keywords:** thymidine kinase 1, TK 210 ELISA, TK-Liaison, CA 125, HE4, ROMA index, ovarian cancer

## Abstract

**Simple Summary:**

Ovarian cancer is one of the most difficult tumors to detect and manage. Usually, it is diagnosed in late stage of the disease which is associated with poor prognosis. Therefore, it is important to detect this cancer in the early stages to improve overall survival. In this study, we determined TK1 protein and TK1 activity levels as well as the biomarkers CA 125, HE4, and the ROMA index. Elevated TK1 protein levels were found in both benign and ovarian tumor (borderline and malignant ovarian cancer) patients. The combination of TK1 protein with CA 125 or HE4 showed higher sensitivity compared to the ROMA index. Therefore, the TK1 protein is a promising serum biomarker that can complement CA 125 or HE4 in the diagnostics of the early stages of ovarian cancer.

**Abstract:**

Background: The early detection of ovarian cancer is presently not effective, and it is crucial to establish biomarkers for the early diagnosis of ovarian cancer to improve the survival of patients. Materials and methods: The aim of this study was to investigate the role of thymidine kinase 1 (TK1) in combination with CA 125 or HE4 to serve as a potential diagnostic biomarkers for ovarian cancer. In this study, a set of 198 serum samples consisting of 134 ovarian tumor patients and 64 healthy age-matched controls were analyzed. The TK1 protein levels in serum samples were determined using the AroCell TK 210 ELISA. Results: A combination of TK1 protein with CA 125 or HE4 showed better performance than either of them alone in the differentiation of early stage ovarian cancer from the healthy control group, but also a significantly better performance than the ROMA index. However, this was not observed using a TK1 activity test in combination with the other markers. Furthermore, the combination of TK1 protein and CA 125 or HE4 could differentiate early stage disease (stage I, II) more efficiently from advanced-stage (stage III, IV) disease (*p* < 0.0001). Conclusions: The combination of TK1 protein with CA 125 or HE4 increased the potential of detecting ovarian cancer at early stages.

## 1. Introduction

Ovarian cancer ranks fifth in cancer deaths among women, accounting for more deaths than any other cancer of the female reproductive system. A woman’s risk of getting ovarian cancer during her lifetime is about 1 in 78. Her lifetime chance of dying from ovarian cancer is about 1 in 108. (These statistics do not count low malignant potential ovarian tumors.). The American Cancer Society estimates that in 2023, about 19.710 new cases of ovarian cancer will be diagnosed and 13.270 women will die of ovarian cancer in the United States [1]. More than two-thirds of ovarian cancer patients are diagnosed in advanced stages of the disease (stage III or IV), which is associated with a 5-year survival of 27% for stage III and 13% for stage IV cancer patients. In contrast, the 5-year survival rate would improve significantly if ovarian cancer was detected in stage I; the 5-year survival in these cases is 90%, and the 10-year survival is 84% [2,3]. There are 1616 women living with ovarian cancer in Slovenia, and the annual observed and estimated incidence of ovarian cancer is 163 cases or 14/100.000 inhabitants. According to the Slovenian Cancer Registry (SCR), the mortality rate is 120, while the one-year survival rate is 75.1 (71.9–78.4) and the five-year survival rate is 43.3 (39.3–47.6) [4]. The pathogenesis of ovarian cancer is not well known; several studies demonstrate different endocrine and genetic factors contributing to the pathogenesis of ovarian cancer [5,6].

The most commonly used serum biomarker for ovarian cancer is CA 125 (Carcinoma antigen 125, also known as mucin 16 or MUC16), which was initially used to monitor ovarian cancer patients during therapy. Earlier studies showed that the detection of ovarian cancer with the combination of CA 125 with pelvic ultrasonography or alone can be used for the large cohort screening of populations [7,8]. However, CA 125 has significant limitations in terms of sensitivity and specificity for the early detection of ovarian cancer in the early stage [9]. Moreover, CA 125 levels are also elevated in different other pathological conditions such as endometriosis and non-malignant gynecologic diseases [10].

In order to improve ovarian cancer detection, other biomarkers such as human epididymis protein 4 (HE4) have been developed [11]. HE4 is a glycoprotein belonging to the family of whey acidic four-disulfide core proteins, and it is overexpressed in serous and endometroid ovarian carcinomas [12,13,14]. Studies have been completed evaluating a series of serum biomarkers for the detection of ovarian carcinomas in women with pelvic masses and the combination of CA 125 with HE4 showed a more efficient prediction of malignancy compared to either of them alone [15].

Furthermore, a dual marker algorithm based on CA 125 and HE4 was developed as a risk of ovarian malignancy algorithm (ROMA) that enhanced the clinical applications of these biomarkers in the differentiation of benign from malignant ovarian carcinomas [14,15,16]. However, another study demonstrated that there was no clinical benefit in using the ROMA index instead of CA 125 or HE4 alone in the detection of epithelial ovarian carcinomas [17]. Since the currently available biomarkers CA 125, HE4, and the ROMA index have these limitations, it is essential to search for an additional biomarker that could improve the sensitivity and specificity of early detection as well as differentiate malignant pelvic masses from benign ovarian tumors.

Thymidine kinase (TK1) is a serum biomarker associated with DNA precursor synthesis which fluctuates during the different phases of the cell cycle making TK1 a unique biomarker for cell proliferation [18,19,20]. Different commercial assays such as TK-Liaison, TK-REA, and Divitum assays are available to measure TK1 activity in sera from a patient with malignant diseases [18]. To enhance the clinical applications of TK1, antibodies were developed against different regions of TK1 which can overcome some of the limitations of TK1 activity assays. Earlier studies showed that TK1 antibody-based dot blot tests had higher sensitivity than TK1 activity based on a diagnosis of patients with solid tumors [21]. AroCell has developed an ELISA-based immunoassay for the quantification of TK1 protein in serum samples. This assay utilizes two monoclonal antibodies developed against the C-terminal region of TK1. Recent studies demonstrate that the AroCell TK 210 ELISA has higher sensitivity compared to TK1 activity assays in the differentiation of patients with solid tumor diseases from healthy individuals [22]. Furthermore, studies demonstrated that TK 210 ELISA could complement tumor-specific biomarkers such as CA 15-3, proPSA, and PHI [23,24].

In this study, we analyzed the diagnostic role of TK1 protein, TK1 activity, CA 125, HE4 as well as the ROMA index for the early detection of ovarian cancer in pre-menopausal and post-menopausal women in comparison with a group of matched controls. Simultaneously, we also evaluated the role of the combination of TK1 with CA 125, HE4, and the ROMA index to identify the best possible combination to improve the diagnostic efficiency for early detection as well as the differentiation of benign ovarian masses from malignant ovarian carcinomas.

## 2. Materials and Methods

### 2.1. Study Population and Sample Collection

This study included 134 serum samples from patients with ovarian tumors; 72 had benign tumors and 62 had borderline and malignant ovarian cancers, and serum samples from 64 healthy women were used as the control group. The serum samples from healthy women and from women with ovarian tumors who attended the Department of Obstetrics and Gynecology, University Medical Centre Ljubljana, were obtained between April 2018 and May 2021. The blood samples were collected from all the patients before surgery and additional information was obtained regarding their lifestyle and gynecological and clinical status. For sample collection, strict standard operating procedures were followed, and serum was aliquoted and stored at −80 °C until analysis. The study was approved by the National Medical Ethics Committee of the Republic of Slovenia (Nr. 109/02/13). All patients gave their written consent for the diagnostic procedures and surgery, as well as inclusion in the study. The data of patients were collected as a prospectively designed database. 

### 2.2. Patient Characteristics 

During the period of 2018 to 2021, a total of 198 (134 patients with ovarian tumors and 64 healthy controls) were included in this study. Among the 134 included women with ovarian tumors, 72 had benign tumors and 62 had borderline and malignant ovarian cancers (Table 1). 

Serous tumors were the most common carcinomas (58%) followed by mucinous tumors (15%) and endometrial tumors (15%). The mean ± SD of age in years was 57.0 ± 3.7 (range = 26.9–85.7) in all patients. Twenty-one patients had a pre-menopausal status in the range of 26.9–50.2 years (mean ± SD = 40.5 ± 3.3), while 41 patients had a post-menopausal status of 52.5–85.7 years (mean ± SD = 65.5 ± 2.9). In the benign ovarian tumor group, 17% were ovarian endometriosis, 15% serous cystadenoma, followed by 13% serous cystadenofibroma, and 10% ovarian mucinous cystadenoma. The patient’s ages were 14.9–85.8 years, with a mean ± SD of 52.01 ± 3.8 years. Of the 72 patients, 33 patients had a pre-menopausal status and their ages were 37.6 ± 3.4, while for the 39 patients with a post-menopausal status, it was 54.3 ± 2.9 years. Of the 64 healthy controls, 42 cases had a pre-menopausal status with a mean± SD of the age of 42.6 ± 2.5 years, and for the 22 cases of post-menopausal status, it was 55.1 ± 2.4 years (Table 1). Histological types of stages of diseases are summarized in Table 2.

### 2.3. Measurement of CA125 and HE4

The serum CA 125 and HE4 levels were determined using in vitro quantitative fully automatic electrochemiluminescent immunoassays (ECLIAs), on a Cobas e411 immunoassay analyzer (Roche Diagnostics GmbH, Manheim, Germany). The method is based on the electrochemiluminescence immunoassay (ECLIA) principle, incorporating a sandwich immunoassay test principle. The serum HE4 and CA125 reference ranges were <140 pmol/L and < 35 kU/L, respectively [25].

### 2.4. The Risk of Ovarian Malignancy Algorithm (ROMA Index)

The pre-menopausal calculation formula of the ROMA index was as follows: PI = −12.0 + (2.38 x LN [HE4]) + (0.0626 x LN [CA-125]) and the calculation for the post-menopausal was: PI = −8.09 + (1.04 x LN [HE4]) + (0.732 x LN [CA 125]). Where LN = Natural Logarithm. The higher risk of ovarian cancer for pre-menopausal women was with a ROMA value of ≥ 11.4% and post-menopausal women had a higher risk with a ROMA value of ≥ 29.9%, as described previously [25].

### 2.5. Serum TK Activity (STK1a) and TK1 Protein (STK1p) Determinations

The TK activity in all the serum samples was analyzed using the LIAISON^®^ assay, as described previously, and the TK activity was expressed in U/L [26]. The TK1 protein levels in serum samples were measured using the Arocell TK 210 ELISA. The TK 210 ELISA is a sandwich-based test that utilizes two monoclonal anti-TK antibodies against the C-terminal region of human TK1. The assay was performed according to the manufacturer’s instructions as previously described and TK1p levels were expressed in ng/mL [22].

### 2.6. Statistical Analysis

The levels of biomarkers including CA 125, HE4, STK1a, and STK1p levels in healthy, benign, and malignant ovarian cancer serum samples were evaluated for normality using the D’Agostino and Pearson omnibus normality test. For continuous variables, the significance of differences was tested using the Mann–Whitney *U* test or Wilcoxon signed-rank test according to the comparison of independent samples or paired samples. Logistic regression analysis was performed to establish the best possible combination of the biomarkers. The diagnostic performance of all possible combinations was evaluated using receiver operating characteristic (ROC) curves and the area under the curve (AUC) with 95% confidence intervals (95%CIs). All statistical analyses were performed using GraphPad Prism 8.0 (GraphPad Software, La Jolla, CA, USA) and MedCalc 17. 6. Statistical significance was achieved when *p* was < 0.05.

## 3. Results

### 3.1. TK1 Levels in the Different Patient Groups

Both serum TK1 protein levels (TK1p) and TK1 activity levels (TK1a) were determined and compared with healthy controls (N = 64), benign (N = 72), and malignant ovarian cancer patients (N = 62) using the AroCell TK 210 ELISA and the TK-Liaison assays. TK1p concentrations were significantly higher in the malignant ovarian cancer group compared to the benign and healthy control groups (*p* < 0.0001; Figure 1A). These results demonstrate that malignant ovarian cancer patients have higher levels of TK1p than patients in the benign and healthy control groups. 

A comparison of TK1p between malignant and benign showed a significant difference between the pre- versus post-menopausal women (Figure 1B,C). For post-menopausal women, TK1p was significantly higher in the malignant compared to benign groups, while there was no significant difference between the malignant and the benign tumor groups in the case of pre-menopausal women (*p* = 0.34). 

The TK1a values showed another type of result with no significant differences between the control group and the malignant ovarian cancer or the benign ovarian tumor groups (Figure 1D). However, the malignant ovarian cancer group had significantly higher TK1a levels compared to a group of benign tumor patients. 

Furthermore, no significant differences were found in the TK1a levels in these groups in pre-menopausal women (Figure 1E). However, significant differences in the TK1a levels were observed between malignant and benign as well as healthy control groups in post-menopausal women (Figure 1F).

In all these groups, CA 125 and HE4 values were determined and the ROMA index values were also calculated as described in materials and methods. The CA 125 levels were significantly higher in the malignant ovarian cancer group compared to the benign group and healthy controls. Furthermore, the CA 125 levels in the malignant group were significantly higher than those in the benign ovarian tumor group. However, both the HE4 and ROMA index values were significantly higher in the malignant ovarian cancer compared to healthy control groups but there was no difference between benign and healthy control groups. Furthermore, there was no significant difference in TK1p levels between pre-menopausal and post-menopausal women in the healthy controls, nor in the case of the benign ovarian tumor and malignant ovarian cancer groups (Table 3). Similar results were observed with TK1a, CA 125, and HE4 in the healthy control groups. The levels of CA 125 in pre-menopausal women were significantly higher compared to those in post-menopausal women in the benign tumor groups. In contrast, the HE4 levels in post-menopausal women were significantly higher than in pre-menopausal women in the benign tumor group (Table 3). In the case of the malignant ovarian cancer group, the TK1a, CA 125, and HE4 levels were significantly higher for post-menopausal than for pre-menopausal women. 

The results of univariate and multivariate ROC curve analyses are shown in Table 4. The ROC area under curves (AUCs) and standard errors were based on the complete data set from 134 ovarian tumor serum samples and 64 healthy controls using TK1p, CA 125, HE4, TK1p + CA 125, TK1p + HE4, and TK1p + ROMA index biomarkers. Multivariate ROC analyses of the combinations of the TK1p, TK1p + CA 125, TK1p + HE4, and TK1p + ROMA index biomarkers were performed (Figure 2A,B). Among all the possible combinations, three dual combinations had an ROC AUC above 0.90. These three combinations were further evaluated for their performance at distinguishing ovarian cancer samples from healthy controls. 

### 3.2. All Ovarian Cancers Versus all Healthy Controls

Figure 2A shows the ROC curves for TK 210 ELISA (TK1p) alone, CA 125 alone, HE4 alone, and the dual markers (TK 210 + CA 125, TK 210 + HE4, and TK 210 + ROMA index) when these combinations were calculated for the 198 samples (134 ovarian tumors and 64 healthy controls). Each biomarker and combination was evaluated for sensitivity, specificity, positive predictive value (PPV), and negative predictive value (NPV) at the ROC curve cut point, with a specificity of around 95%.

Table 5 shows the results of the TK 210 ELISA in combination with CA 125, HE4, and the ROMA index attained a sensitivity above 70% with a specificity of 95%. 

Figure 2B shows the ROC curves for TK-Liaison (TK1a) alone, CA 125 alone, and HE4 alone and the dual markers (TK Liaison + CA 125, TK Liaison + HE4, and TK Liaison + ROMA index). Overall, the TK-Liaison and the combination of TK-Liaison with other assays showed lower sensitivity compared to the TK 210 ELISA assay. However, the combination of CA 125 with either the TK 210 ELISA or TK-Liaison assay showed a higher sensitivity and PPV than either of these markers alone (Table 5).

Based on these results, further analysis was carried out using the TK 210 ELISA, TK 210 + CA 125, TK 210 + HE4, and TK 210 + ROMA index. The patient sera were subgrouped as all pre-menopausal, all post-menopausal women, benign and malignant tumors. The assay performances were evaluated for these subgroups using the ROC curve analysis as shown in Table 6.

### 3.3. Performance Evaluation of Dual Biomarkers in Sub-Groups of Ovarian Cancer Patients 

The performance of the combination of biomarkers with the TK 210 ELISA was evaluated in subgroups of ovarian cancer patients using ROC curve analysis. The diagnostic performances of the TK 210 ELISA alone, TK 210 + CA 125, TK 210 + HE4, and TK 210 + ROMA index are shown in Figure 3 and Table 6. In the detection of all stages of cancers, the AUC for TK 210 + CA 125 was 0.945 (95%CI = 0.879–0.981) for pre-menopausal women (Figure 3A) which is higher than that for TK 210 + ROMA index (AUC = 0.877; 95%CI = 0.795–0.936), and the AUC of TK 210 + ROMA index (AUC = 0.936; 95%CI = 0.87–0.97) is higher than that of TK 210 + CA 125 (AUC = 0.920; 95%CI = 0.85–0.965) for post-menopausal women (Figure 3B). In the detection of benign mass from healthy controls, the AUC for TK 210 + CA 125 (AUC = 0.914; 95%CI = 0.85–0.95) was higher than that for TK 210 + HE4 (AUC = 0.877; 95%CI = 0.81–0.927) and TK 210 + ROMA index (AUC = 0.876; 95%CI = 0.809–0.926; Figure 3C). For the differentiation of malignant ovarian cancer from healthy controls, all three dual biomarkers, i.e., TK 210 + CA 125, TK 210 + HE4, and TK 210 + ROMA index showed higher AUCs compared to the TK 210 ELISA alone (Table 6, and Figure 3D). In the differentiation of benign from malignant ovarian cancer, the AUC was significantly higher for the TK 210 + ROMA index (AUC = 0.819; 95%CI = 0743–0.88) followed by TK 210 + HE4 (AUC = 0.817; 95%CI = 0.741–0.878) and TK 210 + CA 125 (AUC = 0.731; 95%CI = 0.648–0.804), compared to the TK 210 ELISA (AUC = 0.60; 95%CI = 0.52–0.685) alone (Table 6, and Figure 3E). These results demonstrate that the combination of TK 210 + CA 125 gives the best diagnostic performance in the detection of early stage ovarian cancer in comparison with the group with benign tumors. For pre-menopausal women, TK 210 + CA 125 showed higher PPV and NPV than the TK 210 + ROMA index (95.7% vs. 95.1%, and 80.0% vs. 72.7%), whereas in the case of post-menopausal women, the PPV values were similar in all three combinations (100%) with a higher NPV for TK 210 + ROMA index (56.4%) followed by TK 210 + CA 125 (51.2%) and TK 210 + HE4 (50.0%). These results strongly indicate that the dual biomarkers (TK 210 + CA 125) had the best diagnostic performance for the detection of benign ovarian tumors as well as malignant ovarian cancers in comparison to the healthy control group. However, in the differentiation of benign ovarian tumors from malignant ovarian cancers, the combination of the TK 210 + ROMA index gave higher sensitivity (66.1%) than other combinations, as shown in Table 6. A similar analysis was performed using TK-Liaison (TK1a) determinations in these subgroups. In this case, TK-Liaison + CA 125 showed the highest sensitivity compared to TK-Liaison + HE4, and TK-Liaison + ROMA index in the differentiation of pre-menopausal women, benign ovarian tumors, and malignant ovarian cancers from healthy controls (Appendix A). For the differentiation of post-menopausal women with cancer, the TK-Liaison + ROMA index showed higher sensitivity compared to the other combinations. These results are similar to the TK 210 ELISA combination, with CA 125 giving the highest sensitivity for the early detection of ovarian cancer. 

Based on these results, further analysis was carried out using TK 210 + CA 125, TK 210 + HE4, TK-Liaison + CA 125, and TK-Liaison + HE4 combinations. The malignant ovarian cancer serum samples were subclassified based on FIGO staging and TK 210 + CA 125 (*p* < 0.0001; Figure 4A), as well as TK-Liaison + CA 125 (*p* = 0.0104; Figure 4B) in stage III + IV patients, and were significantly higher compared to stage I + II patients. Furthermore, ROC analysis showed that TK 210 + CA 125 (AUC = 0.81, sensitivity = 50% with a specificity = 92%) had higher capacity to differentiate stage I + II from stage III + IV than TK-Liaison + CA 125 (AUC = 0.69; sensitivity = 35.7% with a specificity = 92%; Figure 4C). Similar results were obtained with the combination of TK 210 + HE4 and TK-Liaison + HE4 (Figure 4D,E), and ROC curve analysis showed an AUC of 0.82 for TK 210 + HE4 and 0.80 for TK-Liaison + HE4 (Figure 4F). 

### 3.4. STK1p and STK1a Levels before and after Surgery

Both benign ovarian tumor and malignant ovarian cancer patients (N = 123) were followed after surgery, and the TK1p levels were significantly reduced after surgery for all (*p* = 0.0002; Figure 5A), and for pre-menopausal (*p* = 0.0014; Figure 5B) as well as post-menopausal women (*p* = 0.021; Figure 5C). Among the 123 patients, a total of 81 patients, accounting for 61%, showed a downward trend for the TK1p levels when compared to the preoperative levels. In contrast, there was no significant difference in TK1a levels between preoperative and postoperative patients with benign ovarian tumors as well as malignant ovarian cancer (Figure 5D–F). 

Furthermore, there was a significant correlation between the ratio of TK1p at diagnosis/TK1p after surgery and the number of days after surgery (Figure 6A; rs = 0.25; *p* = 0.0072). In addition, the TK1p was measured in 23 malignant ovarian cancer patients after chemotherapy. With the exception of four patients, the remaining patients showed a significant decrease in TK1p levels after chemotherapy (*p* = 0.013; Figure 6B). During follow-up, 9 patients out of 32 had a relapse of the disease. The patients with relapse after chemotherapy had significantly higher levels of TK 210 + CA 125 as well as TK 210 + HE4 compared to patients without relapse (Figure 6C,D). Similar results were observed with the TK-Liaison determinations as shown in Figure 6E,F.

## 4. Discussion

The success of ovarian cancer treatment is highly dependent on the time of detection of the disease, as patients with an early stage cancer have the greatest chance of survival. Studies have shown that only <25% of patients with ovarian cancer can be diagnosed at the early stage when symptoms are not present, and 70% of patients are diagnosed at the advanced stage. Despite considerable efforts aimed at early detection, no cost-effective screening test has so far been developed [27]. Early detection with the help of tumor-specific biomarkers could thus improve the clinical outcome of patients with ovarian cancer. This is especially important for patients who have vague or no symptoms. In 2008, Moore et al. set up a mathematical algorithm for determining the risk of ovarian cancer (ROMA), which depends on the menopausal status of the woman and the preoperative levels of HE4 and CA 125 in serum [12]. The ROMA index was thus designed with the aim of improving the usefulness of the tumor marker CA 125 in the diagnosis and monitoring of epithelial ovarian cancer.

This study showed that by combining the TK 210 with CA 125 or HE4, one can improve sensitivity for the detection of the early stage of ovarian cancer. This has been also reported in a published study by Cheng Zhu et al. in 2022 [25]. Currently, the use of CA 125 for the detection of ovarian cancer gives relatively low sensitivity and specificity. In this study, we found that the sensitivity and specificity of the ROMA index were 62.9% and 95.3%, in concordance with earlier published studies [12,28,29]. Still, this is not sufficient for the efficient early detection of ovarian cancer. Thus, there is a need for additional biomarkers and the results presented here strongly indicate that TK1 could be such a new diagnostic tool. The addition of TK1p to CA 125, HE4, and the ROMA index demonstrated a sensitivity above 70% with a specificity of 95%. Here, we also found that the sensitivity and specificity of the ROMA index were higher compared to the sensitivity and specificity of CA 125 and HE4 alone [12,25]. In addition, the sensitivity and specificity of TK1a were lower compared to TK1p; 27.6% and 95.3% vs. 62.9% and 95.3%, respectively. However, the combination of TK1a with three markers (CA 125, HE4, and the ROMA index) showed higher sensitivity and specificity than the detection of CA 125, HE4, or the ROMA index alone. We also observed a difference between pre-menopausal and post-menopausal women, in the detection of all stages of cancers. The AUC for TK1p + CA 125 was 0.945 for pre-menopausal women, which was higher than the TK1p + ROMA index (0.877). Furthermore, TK1p + CA 125 showed higher PPV and NPV than TK1p + ROMA index (95.7% vs. 95.1% and 80.0% vs. 72.7%), whereas in the case of post-menopausal women, the AUC of TK1p + ROMA index (0.936) was higher than TK1p + CA 125 (0.920), and the PPV value was similar to all three combinations (100%) with a higher NPV for TK1p + ROMA index (56.4%) followed by TK1p + CA 125 (51.2%) and TK1p + HE4 (50.0%). In the differentiation of benign mass from healthy controls, TK1p + CA 125 had the highest sensitivity (64%) compared to the other combinations.

Our results indicate that higher sensitivity of combinations (TK1p + CA 125 or TK1p + HE4), along with 95% specificity, could offer the early detection of ovarian cancer and therefore improve the prognosis of patients. In the differentiation of malignant ovarian cancer from healthy controls, all three dual biomarkers (TK1p + CA 125, TK1p + HE4, and TK1p + ROMA index) showed higher sensitivity compared to TK1p alone. In the differentiation of benign ovarian tumors from malignant ovarian cancer, the combination of TK1p + ROMA index had a higher sensitivity of 66% followed by TK1p + HE4 at 58% and TK1p + CA 125 at 43.5%, compared to TK1p at 28% alone. These results indicate that the combination of TK1p + CA 125 offers better diagnostic performance in the detection of early stage ovarian cancer, and the TK1p + ROMA index offers higher sensitivity for the differentiation of benign ovarian tumors from malignant ovarian cancer. A similar analysis was performed using TK-Liaison (TK1a) in these subgroups, where TK1a + CA 125 showed higher sensitivity than TK1a + HE4 and TK1a + ROMA index in the differentiation of pre-menopausal women, benign ovarian tumor, and malignant ovarian cancers from healthy controls. The TK1a levels in women with benign ovarian tumors were very low compared to the healthy group but surprisingly the TK1p levels were significantly higher in those with benign masses. Earlier studies have shown that the TK1 undergoes modifications resulting in the inactivation of a substantial fraction of the TK1 protein molecules, especially in low-molecular-weight forms (200–50 kDa) [30]. In addition, in the study of the specific activity of serum, TK1 was two-fold lower in the solid tumor group compared to the healthy group. Moreover, in this study, a similar difference in the median TK1 specific activity was also observed. Further studies need to be performed to clarify the extent and mechanisms of this difference in the specific activities of serum TK1 in healthy persons and those with cancer diseases. These variations in TK1-specific activity further substantiate the fact about the inactivation of TK1 in solid tumors. All these factors could possibly explain the reason for low TK1a and high TK1p in benign masses. However, the VEGF levels were significantly higher in malignant ovarian cancer indicating angiogenesis and more new blood vessel formation that may facilitate the transfer of TK1 into the main bloodstream, which provides an explanation for the higher levels of TK1a and TK1p in malignant ovarian cancer [31].

In our study, serum TK1p was statistically significant in the ovarian cancer group (*p* < 0.0001) as compared to those in the benign group as well as the control group. In addition, a combination of TK1p with CA 125, as well as HE4, significantly differentiates ovarian cancer patients based on stage, and identifies the probability of patients with tumor relapse. Additional validation studies using the combination of TK1p with CA 125 or HE4 for the early detection of ovarian cancer patients are warranted.

## 5. Conclusions

The results of this study showed that the combination of TK1p with CA 125 or HE4 could improve the sensitivity and specificity of the early detection of ovarian cancer. The combination of these biomarkers may offer a route to improving the detection of patients with ovarian cancer in the early stages of the disease leading to a higher chance of curative treatment. Future larger clinical studies should aim at optimizing the combinations of TK1p with CA 125 or HE4 to establish diagnostic standards for ovarian cancer screening.

## Figures and Tables

**Figure 1 cancers-15-01593-f001:**
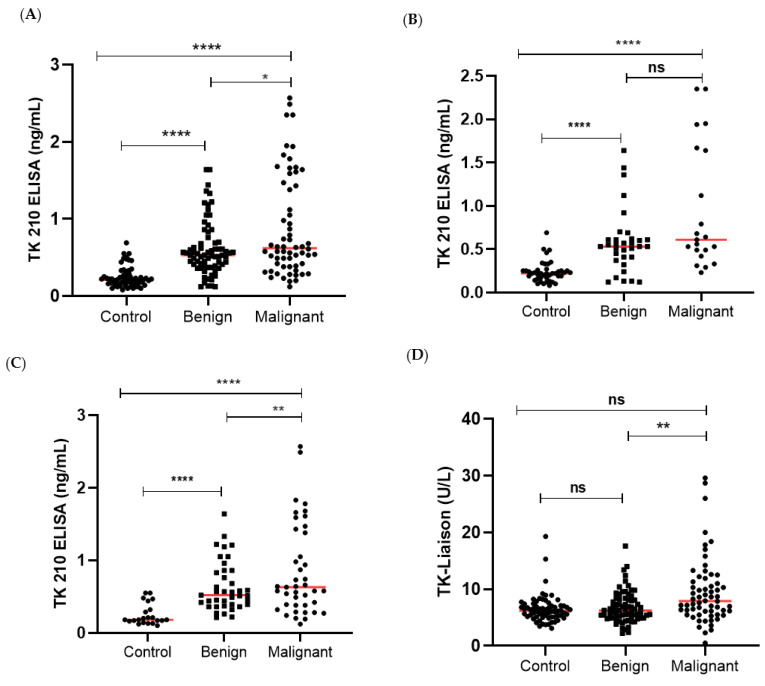
(**A**) TK1 210 (TK1p) in healthy controls, benign ovarian tumors, and malignant ovarian cancer patients. (**B**,**C**) TK1 210 (TK1p) levels between control, benign, and malignant group in the pre- versus post-menopausal women. (**D)** TK1-Liaison (TK1a) in healthy controls, benign ovarian tumors, and malignant ovarian cancer patients. (**E**,**F**) TK1-Liaison (TK1a) levels between control, benign, and malignant group in the pre- versus post-menopausal women (ns *p* > 0.05, * *p* ≤ 0.05, ** *p* ≤ 0.01, **** *p* ≤ 0.0001).

**Figure 2 cancers-15-01593-f002:**
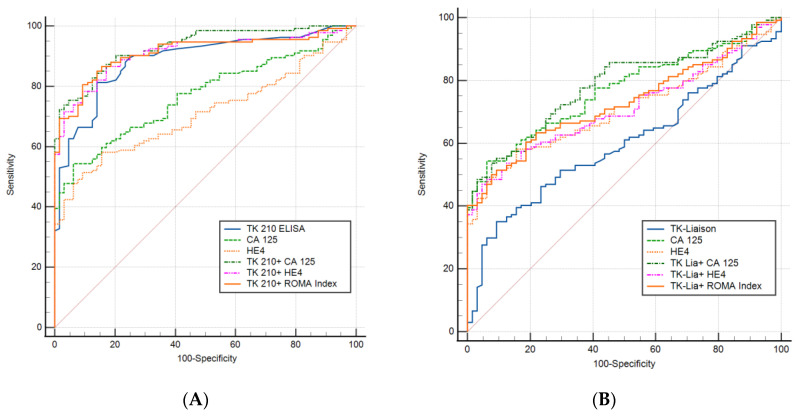
(**A**,**B**) ROC curves for TK 210 ELISA, TK-Liaison, CA 125, and HE4 alone, and the dual markers with TK 210 ELISA and TK-Liaison.

**Figure 3 cancers-15-01593-f003:**
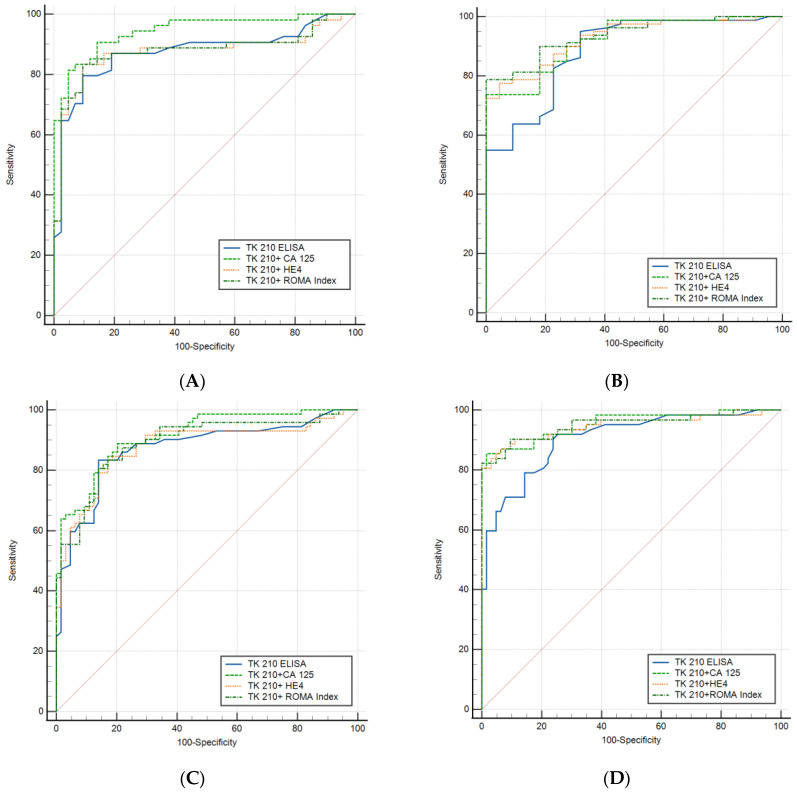
(**A**–**E**) The diagnostic performances of the TK 210 ELISA, TK 210 + CA 125, TK 210 + HE4, and TK 210 + ROMA index.

**Figure 4 cancers-15-01593-f004:**
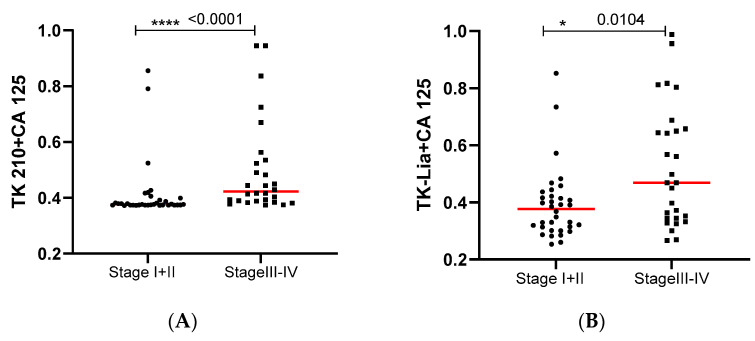
(**A**–**F**) The combination of TK 210 + CA 125, TK-Liaison + CA 125, TK 210 + HE4, TK-Liaison + HE4, and analysis of the ROC curve (* *p* ≤ 0.05, **** *p* ≤ 0.0001).

**Figure 5 cancers-15-01593-f005:**
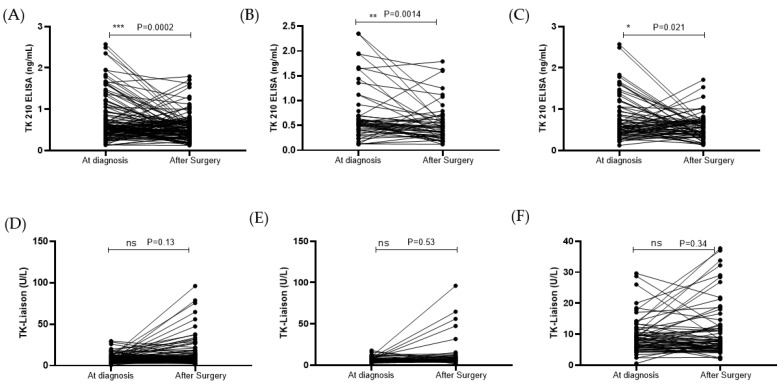
(**A**–**F**) STK1p and STK1a levels before and after surgery (ns *p* > 0.05, * *p* ≤ 0.05, ** *p* ≤ 0.01, *** *p* ≤ 0.001).

**Figure 6 cancers-15-01593-f006:**
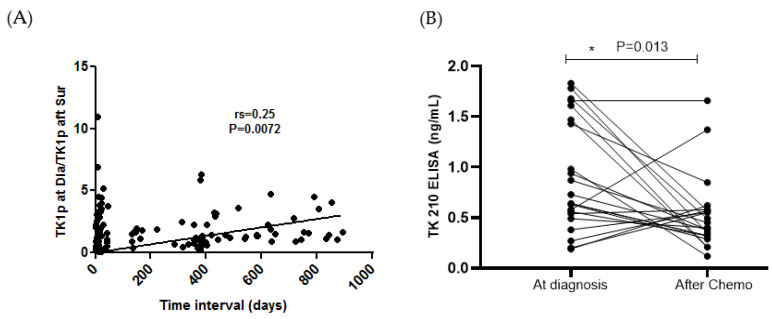
(**A**–**F**) The ratio of TK1p at diagnosis/TK1p after surgery and with the number of days, TK1p levels after chemotherapy for levels of TK 210 + CA 125 as well as TK 210 + HE4, TK-Liaison levels after chemotherapy for levels of TK 210 + CA 125 as well as TK 210 + HE4 (* *p* ≤ 0.05).

**Table 1 cancers-15-01593-t001:** Detailed clinical characteristics of the study participants.

Characteristics	N/%	Mean ± SD (ys)
Ovarian tumor (borderline and malignant ovarian cancer)	62	57.0 ± 3.7
Menopausal status		
Pre-menopause	21	40.5 ± 3.3
Post-menopause	41	65.5 ± 2.9
Histologic type		
Serous carcinomas	58%	
Mucinous tumors	15%	
Endometrial carcinomas	15%	
Benign ovarian tumor	72	52.01 ± 3.8
Menopausal status		
Pre-menopause	33	37.6 ± 3.4
Post-menopause	39	54.3 ± 2.9
Histologic type		
Endometriosis	17%	
Serous cystadenoma	15%	
Serous systadenofibroma	13%	
Mucinous cystadenoma	10%	
Healthy controls	64	47.2 ± 2.3
Menopausal status		
Pre-menopause	42	42.6 ± 2.5
Post-menopause	22	55.1 ± 2.4

**Table 2 cancers-15-01593-t002:** Histological types and distribution of stages of diseases for all patients as well as the distribution of the pre-menopausal and post-menopausal women.

Histopathology and Classification	All Patients (N)	Menopausal Status	
	134	pre-(N)	post-(N)
Benign ovarian tumor	72	33	39
Serous cystadenoma	11	2	9
Serous cystadenofibroma	9	1	8
Mucinous cystadenoma	7	4	3
Mucinous cystadenofibroma	2	0	2
Endometriosis	12	12	0
Benign Brenner tumor	1	0	1
Sclerosing stromal tumor	1	1	0
Mature teratoma	8	5	3
Follicle cyst	2	2	0
Corpus luteum cyst	1	1	0
Inclusion cyst	7	2	5
Simple cyst	9	3	6
Cellular fibroma	2	0	2
Ovarian tumor (borderline and malignant ovarian cancer)	62	21	41
Serous borderline tumor	9	6	3
Low-grade serous carcinoma	4	1	3
High-grade serous carcinoma	17	3	14
Mucinous borderline tumor	7	5	2
Mucinous carcinoma	2	1	1
Endometrioid carcinoma	8	2	6
Endometrioid borderline tumor	1	0	1
Clear cell carcinoma	2	1	1
Seromucinous borderline tumor	3	0	3
Adult granulosa cell tumor	2	1	1
Dysgerminoma	1	1	0
High-grade primary peritoneal serous carcinoma	5	0	5
Low-grade primary peritoneal Serous carcinoma	1	0	1
Healthy controls	65	43	22
FIGO stages	62		
I	29	13	16
II	5	2	3
III	24	6	18
IV	4	0	4
Grade	62		
G1	35	16	19
G2	6	3	3
G3	21	2	19

**Table 3 cancers-15-01593-t003:** TK 210 ELISA, TK-Liaison, CA 125, and HE4 levels in different groups.

	N	TK1p (ng/mL)	*p* Value	TK1a (U/L)	*p* Value	CA125 (kU/L)	*p* Value	HE4 (pmol/L)	*p* Value
Healthy controls									
All	64	0.21 (0.16–0.27)		6.22 (5.1–7.1)		12.16 (9.77–17.7)		47.2 (42.1–54.9)	
Pre-menopausal	42	0.22 (0.167–0.25)		6.07 (5.1–6.9)		13.17 (10.3–19.8)		45.4 (38.2–56.9)	
Post-menopausal	22	0.18 (0.155–0.35)	0.625	6.6 (5.3–7.5)	0.299	10.66 (8.8–16.1)	0.113	49.7 (44.5–53.5)	0.429
Benign masses									
All	72	0.53 (0.39–0.70)		6.62 (5.0–8.2)		17.5 (10.5–41.2)		50 (42.0–66.4)	
Pre-menopausal	33	0.53 (0.39–0.62)		6.43 (4.7–7.7)		31.9 (15.2–69.1)		46.8 (39.3–59.0)	
Post-menopausal	39	0.52 (0.39–0.86)	0.68	5.99 (4.9–8.8)	0.688	13.9 (9.0–25.2)	0.0005	57.8 (44.7–75.6)	0.0088
Malignant masses									
All	62	0.62 (0.38–1.44)		7.91 (5.95–11.4)		103.8 (21.4–444)		136.5 (62.4–417.8)	
Pre-menopausal	21	0.61 (0.45–1.65)		6.52 (5.4–9.1)		42.7 (19.3–193)		52.0 (40.5–128.6)	
Post-menopausal	41	0.63 (0.38–1.40)	0.64	9.02 (6.3–12.8)	0.045	169.2 (27.6–700)	0.047	290.8 (93–972)	< 0.0001

**Table 4 cancers-15-01593-t004:** Univariate and multivariate ROC analysis of biomarkers.

Marker Name	ROC AUC ± SE (AUC)
TK1p	0.88 ± 0.024
TK1a	0.60 ± 0.04
CA 125	0.77 ± 0.032
HE4	0.71 ± 0.035
ROMA Index	0.73 ± 0.034
TK1p + TK1a	0.89 ± 0.024
TK1p + CA 125	0.93 ± 0.020
TK1p + HE4	0.88 ± 0.024
TK1p + CA125 + HE4	0.94 ± 0.015
TK1p + ROMA Index	0.91 ± 0.020
TK1a + CA 125	0.78 ± 0.032
TK1a + HE4	0.71 ± 0.035
TK1a + CA125 + HE4	0.80 ± 0.030
TK1a + ROMA Index	0.73 ± 0.034

**Table 5 cancers-15-01593-t005:** ROC curve analysis for the different biomarkers alone, and in combination.

Biomarker	Cut-Off	AUC	Sensitivity	Specificity	PPV	NPV
TK 210 ELISA	0.50	0.88	62.9%	95.3%	96.6%	55.1%
TK-Liaison	9.10	0.598	27.6%	95.3%	90.2%	38.2%
CA 125	26.4	0.77	54.48%	93.75%	94.8%	49.6%
HE4	65.5	0.71	47.8%	93.75%	94.1%	46.2%
ROMA Index	16.7	0.725	42.54%	95.31%	95.0%	44.2%
TK 210 + CA 125	0.78	0.932	75.3%	95.3%	97.1%	64.9%
TK 210 + HE4	0.80	0.91	73.8%	93.75%	96.1%	63.2%
TK 210 + ROMA Index	0.83	0.912	70.2%	95.3%	96.9%	60.4%
TK-Liason + CA 125	0.76	0.783	49.25%	95.3%	95.7%	47.3%
TK-Liason + HE4	0.72	0.712	47.1%	95.3%	95.5%	46.2%
TK-Liason + ROMA Index	0.73	0.728	44.1%	95.3%	95.2%	44.9%

**Table 6 cancers-15-01593-t006:** ROC curve analysis of TK 210 ELISA in combination with other biomarkers in different subgroups.

Pre-Menopausal vs. Controls	Cut-Off	AUC	Sensitivity	Specificity	PPV	NPV
All stages						
TK 210 ELISA	0.49	0.875	64.8%	95.2%	94.6%	67.80%
TK 210 + CA 125	0.68	0.945	81.5%	95.2%	95.7%	80.0%
TK 210 + HE4	0.75	0.875	72.2%	95.2%	95.1%	72.7%
TK 210 + ROMA Index	0.75	0.877	72.2%	95.2%	95.1%	72.7%
Post-menopausal vs. controls						
All stages						
TK 210 ELISA	0.55	0.886	55.0%	100%	100%	37.9%
TK 210 + CA 125	0.86	0.920	73.75%	100%	100%	51.2%
TK 210 + HE4	0.86	0.927	72.5%	100%	100%	50.0%
TK 210 + ROMA Index	0.83	0.936	78.75%	100%	100%	56.4%
Benign vs. controls						
TK 210 ELISA	0.50	0.872	59.72%	95.3%	93.5%	68.1%
TK 210 + CA 125	0.76	0.91	63.89%	96.87%	95.9%	71.3%
TK 210 + HE4	0.77	0.877	61.1%	95.3%	93.6%	68.5%
TK 210 + ROMA Index	0.79	0.876	59.72%	95.3%	93.5%	67.8%
Malignant vs. controls						
TK 210 ELISA	0.50	0.904	66.13%	95.3%	93.2%	74.7%
TK 210 + CA 125	0.65	0.954	80.65%	100%	100%	84.2%
TK 210 + HE4	0.577	0.947	80.65	100%	100%	84.2%
TK 210 + ROMA Index	0.606	0.951	82.26%	100%	100%	85.1%
Malignant vs. Benign						
TK 210 ELISA	1.36	0.61	27.5%	95.8%	85%	60.5%
TK 210 + CA 125	0.56	0.73	43.5%	95.8%	90%	66.3%
TK 210 + HE4	0.60	0.817	58.1%	97.22%	94.7%	72.9%
TK 210 + ROMA Index	0.59	0.819	66.1%	95.8%	93.2%	76.7%

## Data Availability

The data that support the findings of the study are available from the corresponding author upon reasonable request.

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
