# Peer review of "A Dual Biomarker TK1 Protein and CA125 or HE4-Based Algorithm as a Better Diagnostic Tool than ROMA Index in Early Detection of Ovarian Cancer"

_cancers, 2023, doi:10.3390/cancers15051593_

Round 1

Reviewer 1 Report

Ovarian cancer is one of the most difficult tumors to detect and manage. It is important to detect this type of cancer in early stages hence to improve the overall survival. If this condition is not fulfilled the prognosis is usually unfavorable. During the time, various tumour markers were tested to help us in early detection. Currently, the group of tumour markers is used in the routine clinical practice to detect ovarian cancer and also additional calculated tumour markers like ROMA indexes were developed. However, the sensitivity of currently used tumour markers is unsatisfied and each contribution to its improvement is appreciated.

The manuscript is well written with the clear methodology. The study group is well defined and contains cancer group, benign tumours and healthy subjects. Thymidine kinase is an additional tumour marker which is used to assess cell division and activity of the tumour. Two types of TK kits with different principles are used to increase sensitivity of the diagnostic process of ovarian cancer. Routinely used tumour markers CA125 and HE4 and ROMA index are determined and calculated the accordingly are added the results of two types of thymidine kinase and the sensitivity of such approaches are compared.

I recommend this article for publication after my following comments are resolved:

1)      Usually the patient group characteristics subsection is a part of the Materials and Methods. I recommend to transfer the whole subsection 3.1. Patient characteristics from 3. Results to
2. Materials and Methods as the subsection 2.2. Patients characteristics.

2)     Line 261 - Unify the notation of the used pairs of kits  - TK Lia+ CA 125, TK Lia+HE4, and TK Lia+ ROMA Index (space or without space)

3)      Lines 264, 267 and 268 - Improve and unify the name of the kit TK-Liaison. TK-Liaison is the right name of the kit. Correct the name of the kit in the line 264, 267 and 268.

4)      Line 450 – Futur – is it correct?

Reviewer 2 Report

In this study, Authors have determined TK1 protein levels and activity in association with the biomarkers CA 125, HE4, and ROMA index in 199 serum samples from 134 ovarian tumor patients and 65 healthy controls. They found that this combination showed higher sensitivity compared to the ROMA index, thus concluding that the TK1 protein is a promising serum biomarker that can improve CA 125 or HE4 diagnosis of ovarian cancer early stages.

The results presented in a quite large number of figures and tables actually seem to confirm the conclusions. However, some points need to be improved, particularly in the way of presenting some figures.

Fig. 1; it would be better to write directly in the figure next to each panel, what they refer to, for a more immediate understanding and not only in the text.

Fig. 2A, 2B; Figure 3; Figure 4C and 4F; please enlarge the inset since in all these figures it is very small and the symbols and writings inside the inset are barely readable.

Why figures 2A and 2B are separate? Isn't it better to merge them into one?

Since serous ovarian carcinoma is the most aggressive type, the extrapolation of data relating to only this tumor type could be further more indicative of the usefulness of this analysis and increase its importance.

The reference Zhu, C.; Zhang, N.; Zhong, A.; Xiao, K.; Lu, R.; Guo, L. A combined strategy of TK1, HE4 and CA125 shows better diagnostic performance than risk of ovarian malignancy algorithm (ROMA) in ovarian carcinoma. Clin. Chim. Acta 2022, 524, 43–50, is mentioned only about methods, but it reports studies with a similar rationale. Nonetheless it is not mentioned and discussed in the Discussion, why? What are the differences between that work and the submitted manuscript?

Reviewer 3 Report

The authors examined the efficacy of TK1 protein and CA125 or HE4-based algorithm as a diagnostic tool for ovarian cancer. The theme of this study is important. The topic of this study is important. However, the reviewer has several concerns about the study design of this study. The reviewer's comments are as follows.

Lines 47-48

Please use the latest statistic data.

Line 150-166

Please show the demographics of the patients. In addition, the difference in patient demographics between the healthy control, benign disease, and malignant disease groups should be determined using statistical analyses.

Table 1

The critical problem of the current study is the severe heterogeneity of histological subtypes in malignant disease.

Results

The results may be statistically significant, but they do not appear to be clinically useful.

Overall, the severe heterogeneity in the malignant disease is the critical problem of this study that can not be addressed. 

Round 2

Reviewer 2 Report

Authors have satisfactorily replied to the questions raised by this reviewer.

Reviewer 3 Report

While I pointed out the problems of current studies, the authors made maximum effort to improve the manuscript. I think the authors improved the manuscript according to my previous comments.